# Supplementation of EGF, IGF-1, and Connexin 37 in IVM Medium Significantly Improved the Maturation of Bovine Oocytes and Vitrification of Their IVF Blastocysts

**DOI:** 10.3390/genes13050805

**Published:** 2022-04-30

**Authors:** Sha Yang, Yuze Yang, Haisheng Hao, Weihua Du, Yunwei Pang, Shanjiang Zhao, Huiying Zou, Huabin Zhu, Peipei Zhang, Xueming Zhao

**Affiliations:** 1Institute of Animal Sciences (IAS), Chinese Academy of Agricultural Sciences (CAAS), Beijing 100193, China; 82101192351@caas.cn (S.Y.); haohaisheng@caas.cn (H.H.); duweihua@caas.cn (W.D.); pangyunwei@caas.cn (Y.P.); zhaoshanjiang@caas.cn (S.Z.); zouhuiying@caas.cn (H.Z.); zhuhuabin@caas.cn (H.Z.); 82101209707@caas.cn (P.Z.); 2Beijing General Station of Animal Husbandry, Beijing 100101, China; yyz84929056@126.com

**Keywords:** IVM, EGF, IGF, connexin 37, vitrification, bovine, oocyte

## Abstract

The quality and developmental capacity of oocytes derived from in vitro maturation (IVM) remain unsatisfactory, which greatly impairs the efficiency and application of embryo technologies. The present experiment was designed to investigate the effect of the supplementation of EGF, IGF-1, and Cx37 in an IVM medium on the maturation quality and development ability of bovine oocytes. The cytoplasmic maturation events of oocytes and the quality of in vitro fertilization (IVF) blastocysts were examined to investigate the relative mechanisms. Our results showed that the nuclear maturation and blastocyst development after the IVF of oocytes treated with 25 μg/mL Cx37 or the combination of 50 ng/mL EGF and 100 ng/mL IGF-1 were significantly increased compared to those of the control group (*p* < 0.05). Furthermore, the blastocyst rate, and blastocyst total cell number and survival rate after vitrification of the EGF+IGF-1+Cx37 group, were significantly higher than those of the control group (*p* < 0.05), but lower than those of the FSH+LH+EGF+IGF-1+Cx37 group (*p* < 0.05). The transzonal projection (TZP) intensity, glutathione (GSH) level, and mitochondrial function of the EGF+IGF-1+Cx37 group were significantly higher than that of the control group, and lower than those of the FSH+LH+EGF+IGF-1+Cx37 group, in contrast to the results of the reactive oxygen species (ROS) levels. In conclusion, our results showed that the supplementation of 50 ng/mL EGF, 100 ng/mL IGF-1, and 25 μg/mL Cx37 in the IVM of bovine oocytes significantly improved their quality and developmental ability by increasing the TZP, mitochondrial function, and GSH level.

## 1. Introduction

In vitro maturation (IVM) is a kind of assisted reproductive technology, which permits immature oocytes collected from the mid-size antral follicles of unstimulated ovaries to be matured in vitro [1]. It provides abundant oocyte resources for agricultural research and biomedical purposes [2], including nuclear transfer and transgenic technology [3]. IVM can be used as an effective tool to study the key factors of early embryonic development [4], preimplantation embryonic development, and post-transfer survival [5]. For patients with infertility problems, IVM can reduce the cost of treatment [6], simplify the treatment process, and avoid the potential of side effects by reducing gonadotrophin stimulation in patients [7,8].

However, going through decades of research, many studies demonstrate that the quality and developmental ability of IVM oocytes were poorer than those matured in vivo [9,10], including goats [11], cattle [9], mice [12], and rabbits [13]. IVM has been shown to cause a significant ultrastructural heterogeneity of oocytes, as well as the desynchronization of nuclear maturation and cytoplasmic maturation [9]. The volume density of mitochondria and the nuclear volume density in in vitro fertilization (IVF) blastocysts are lower than those in blastocysts obtained in vivo, in contrast to the lipid volume density [14]. Furthermore, IVM blastocysts showed significantly lower cumulative biochemical pregnancy, clinical pregnancy, and live-birth rates [15].

In vivo, cumulus–oocyte complexes (COCs) are bathed with follicular fluid, which contains a series of proteins, cytokines, hormones, energy metabolites, steroids, and some undefined factors [16]. However, a single non-physiological IVM system has now been utilized in the IVM of oocytes [17]. Researchers have added different factors in the IVM medium to improve the maturation quality of oocytes. EGF is the founding member of the EGF ligand family, and it stimulates oocyte maturation and cumulus expansion [18]. IGF-1 is an effective mitogen of granulosa cells [19], and it enhances the nuclear maturation of bovine [20], cat [21], and zebrafish [22] oocytes. Connexin37 (Cx37) is a member of the connexin family that contributes to the gap junction construction [23], enhances the communication between granulosa cells and oocytes, and facilitates the maturation of oocytes [24]. However, little information is available about the effect of the combined treatment of EGF, IGF-1, and Cx37 on the maturation of bovine oocytes.

To support the maturation of oocytes, most IVM media contain higher concentrations of FSH than those present in the follicular fluid where follicles eventually grow and ovulate [25]. However, high doses of FSH may affect oocyte chromatin dynamics and oocyte transcription/translation activities, impel gap connection-mediated communication between cumulus cells and oocytes by reducing transzonal projection (TZP) density [26], and impair oocyte development and embryo production [27]. Moreover, the high concentration of LH will significantly decrease polar body formation, initial cleavage, and blastocyst development [28]. Until now, many experiments have been designed to culture immature oocytes in an IVM medium without LH and FSH [29,30].

Therefore, the present experiment was designed to investigate the effect of the supplementation of EGF, IGF-1, and Cx37 in an IVM medium on the maturation and development ability of bovine oocytes. Furthermore, the addition of EGF, IGF-1, and Cx37 in an IVM medium without gonadotropin was also examined. Finally, the cytoplasmic maturation events (TZP, glutathione (GSH), and mitochondrial function) of oocytes and the quality of IVF blastocysts were examined to investigate the mechanisms via which EGF, IGF-1, and Cx37 worked.

## 2. Materials and Methods

In the present study, all chemicals were purchased from Sigma-Aldrich Chemical Company (Missouri, MO, USA), and plastic products were obtained from Thermo Fisher Scientific Company (Massachusetts, MA, USA). All animal processing was premised based on the Institutional Animal Care and Use Committee of the Chinese Academy of Agricultural Sciences.

### 2.1. IVM of Bovine Oocytes

Bovine ovaries were collected from local slaughterhouse, and transported to the laboratory within 2 h in a physiological saline solution with penicillin and streptomycin. Follicles with a diameter of 2–8 mm were selected to collect cumulus–oocyte complexes (COCs), and only those with more than 3 layers of compact cumulus cells were used for the experiment. For IVM, 50 COCs were incubated in 750 μL IVM medium in each well with 5% CO_2_ at 38.5 °C for 22–24 h. The basic IVM medium contained medium 199 (Gibco BRL, Carlsbad, CA, USA) supplemented with 10% (*v/v*) fetal bovine serum (FBS, Hyclone; Gibco BRL) and 10 μg/mL estradiol.

According the experiment design, the oocytes were cultured in different IVM medium as below.

For the control group, the IVM medium contained medium 199 (Gibco BRL, Carlsbad, CA, USA) supplemented with 10% (*v/v*) fetal bovine serum (FBS, Hyclone; Gibco BRL), 5 IU/mL FSH, 10 IU/mL LH, and 10 μg/mL estradiol.

In the first experiment, 100 ng/mL IGF-1 or 50 ng/mL EGF were added into the basic IVM medium of bovine COCs in the presence of FSH and LH, and the maturation and development ability of bovine oocytes were examined.

In the second experiment, 100 ng/mL IGF-1 or 50 ng/mL EGF were added into the basic IVM medium of bovine COCs without FSH and LH, and the maturation and development ability of bovine oocytes were examined.

In the third experiment, different concentrations of Cx37 (0, 12.5 μg/mL, 25 μg/mL, and 50 μg/mL) were added to the basic IVM medium of bovine COCs in the presence of FSH and LH, and nuclear maturation and development, and the optimal Cx37 concentration, were examined.

Finally, 100 ng/mL IGF-1, 50 ng/mL EGF, and 25 μg/mL Cx37 were added together into the basic IVM medium of oocytes with or without FSH and LH, and cytoplasmic events of bovine oocytes and the quality of their IVF blastocysts were examined.

### 2.2. IVF of Oocytes

The IVF procedure was performed according to the method described by Brackett and Oliphant [31]. After being thawed in 38 °C water, one straw of frozen semen (Beijing Dairy Cattle Center, Beijing, China) was washed twice in 7 mL Brackett and Oliphant (BO) medium by centrifugation at 1500 rpm for 5 min. Then, the sperm pellet was resuspended in fertilization medium (BO medium containing 20 μg/mL heparin), and 10 μL sperm suspension was added to 90 μL fertilization medium to make the final sperm concentration of 1 × 10^6^/mL. Subsequently, 20 to 30 oocytes were placed in 100 μL fertilization droplet. After 16–18 h of fertilization, the presumed zygotes were washed and cultured in Charles Rosenkrans medium [32] for 48 h. Finally, the cleavage embryos were cultured in Charles Rosenkrans medium with 10% FBS for another 120 h to blastocysts.

### 2.3. qRT-PCR of Candidate Genes in Oocytes and Blastocysts

As shown in Table 1, the qRT-PCR procedure was performed on an ABI 7500 SDS instrument (Applied Biosystems, Foster City, CA, USA) using the comparative Ct (2^−ΔΔCt^) method, and β-ACTIN was used as a reference gene. Each reaction was performed in a total volume of 25 µL, and the PCR program consisted of a denaturing cycle (95 °C for 2 min) and 40 cycles of PCR (95 °C for 10 s, 60 °C for 30 s) [33].

### 2.4. Examination of Total Nuclear Cell per Blastocyst

Blastocysts were treated with 5 mg/mL pronase to remove zona pellucida, then stained with 10 μg/mL of Hoechst-33342 for 10 min. Finally, they were washed, mounted, and examined under a fluorescence microscope (Olympus, Notting Hill, Australia) equipped with a CoolSNAP HQ CCD camera.

### 2.5. Vitrification of Blastocysts

The vitrification procedure of IVF blastocysts was performed according to the method described by Zhao et al. [34]. For vitrification, embryos were incubated in 10% EG + 10% DMSO for 30 s, treated with EDFS30 for 25 s, and finally plunged into liquid nitrogen (LN_2_) in open-pulled straws (OPS). For warming, embryos were expelled from the OPS and incubated in 0.5 M sucrose for 5 min. Then, embryos were incubated in CR1aa for 30 min and those that survived were morphologically evaluated.

### 2.6. Analysis of TZP Intensity in Oocytes

The immunofluorescence staining of TZP in oocytes was performed according to the methodology described by Yuan et al. [35]. Firstly, oocytes were fixed in 4% paraformaldehyde for 30 min at room temperature. Then, oocytes were permeated in 0.1% Triton X-100 for 5 min and blocked in 1% BSA for 1 h. Finally, oocytes were incubated in Rhodamine Phalloidin (ab235138, 1:1000 dilution; Abcam, Cambridge, UK) for 1 h and imaged with confocal microscopy (TCS SP8; Lecia Leica Microsystems, Wetzlar, Germany). The fluorescent intensity of TZP staining was analyzed with ImageJ software (Version1.40; National Institutes of Health, Bethesda, MD, USA).

### 2.7. GSH Assay in Oocytes

Intra-oocyte GSH content was measured as the method previously described in reference [36]. A group of 20–30 oocytes was transferred into microtube containing 5 µL of 0.2 M sodium phosphate buffer with 10 mM Na2-EDTA and 5 µL of 1.25 M phosphoric acid. Then, 350 µL of 0.33 mg/mL NADPH, 50 µL of 6 mM DTNB, and 90 µL of distilled water was added into the microtube and mixed. Finally, 5 μL of 250 U/mL GSH reductase was added to the microtube to start the reaction. Absorbance was monitored at 412 nm using a spectrophotometer for 3 min, and total GSH content in oocytes was calculated from a constructed standard curve [36].

### 2.8. Analysis of ROS Level in Oocytes

The ROS levels in oocytes were detected using the method described by Rahimi et al. [37] with modifications. A fluorescent dye, 2′,7′-dichlorodihydrofluorescein diacetate H2DCF-DA; Genmed Scientific Inc, Wilmington, DE, USA), was used to measure intra-cellular redox state. Briefly, oocytes were transferred to working solution containing dye solution and incubated for 20 min at 37 C° in 5% CO_2_. Then, oocytes were transferred into 96-well dishes with 100 μL storage solution in each well, and luminescence was immediately measured using a luminometer (InfiniteM200, Tecan Group Ltd., Männedorf, Switzerland). The fluorescence excitation level and emission level were set at 488 nm, and 530 nm, respectively, and ROS values were expressed as counted photons per sec (c.p.s).

### 2.9. ATP Content of Oocytes

ATP content was determined quantitatively by measuring the luminescence generated in an ATP-dependent luciferin–luciferase bioluminescence assay (ATP Bioluminescence Assay Kit HS II, Roche Diagnostics GmbH, Mannheim, Germany), as described by Van Blerkom et al. [38]. Briefly, an aliquot (20 μL) of cell lysis reagent was added to a 0.5 mL centrifuge tube containing oocytes, and the oocytes were homogenized and lysed by repeated pipetting. A standard curve containing ATP concentrations from 0.01 to 10.0 pmol was generated for each series of analyses. ATP detection solution (100 μL) was added to 96-well dishes and equilibrated for 3–5 min. Standard solutions and samples (20 μL) were added to each well. Luminescence was immediately measured using a luminometer (InfiniteM200, Tecan Group Ltd., Untersbergstrasse, Austria) for 10 s. A standard curve was generated from the relative light intensity of the serial dilutions, and ATP content of the samples was determined from the standard curve.

### 2.10. Statistical Analysis

Results were expressed as means ± standard error, and all experiments were repeated at least 3 times. All data were analyzed using a one-way analysis of variance with Statistical Analysis System software 8.1 (SAS Institute, Raleigh, NC, USA), and *p* < 0.05 was considered statistically significant.

## 3. Results

### 3.1. Effect of EGF and IGF-1 on the Maturation and Developmental Ability of Bovine Oocytes

As shown in Table 2, the nuclear maturation rate of the FSH+LH+EGF group was similar to that of the FSH+LH+IGF-1 group and control group (*p* > 0.05), but significantly lower than that of the FSH+LH+EGF+IGF-1 group (*p* < 0.05). Meanwhile, when oocytes were used for IVF, the cleavage rate and blastocysts development rate of the FSH+LH+EGF group and FSH+LH+IGF-1 group were similar to those of the control group (*p* > 0.05), but significantly lower than those of the FSH+LH+EGF+ IGF-1 group (*p* < 0.05).

### 3.2. Effect of EGF and IGF-1 on the Maturation and Developmental Ability of Bovine Oocytes without Gonadotropins

As shown in Table 3, the nuclear maturation rate of the EGF+IGF-1 group was similar to that of the control group (*p* > 0.05), but significantly higher than that of the EGF group, IGF-1 group, and -FSH-LH group (*p* < 0.05). At the same time, when the oocytes were used for IVF, the cleavage rate and blastocysts development rate of the EGF+IGF-1 group were similar to those of the control group (*p* > 0.05), but significantly higher than those of the EGF group, IGF-1 group, and -FSH-LH group (*p* < 0.05).

### 3.3. Effect of Cx37 on the Maturation and Developmental Ability of Bovine Oocytes

To explore the Cx37 on the maturation of oocytes, three concentrations of Cx37 were used in the IVM of bovine COCs. As shown in Table 4, there was no significant difference in the nuclear maturation rates among the four groups. However, the cleavage rate and blastocysts development rate of the 25 μg/mL Cx37 group were significantly higher than the 12.5 μg/mL Cx37 group, 50 μg/mL Cx37 group, and the control group.

### 3.4. Effect of the Combination Treatment of EGF, IGF-1, and Cx37 on TZPs in Bovine Oocytes

Figure 1A shows the representative image of TZPs staining of the bovine oocytes. As shown in Figure 1B, the fluorescence intensity of the EGF+IGF-1+Cx37 group was significantly higher than that of the control group (*p* < 0.05), and lower than that of FSH+LH+EGF+IGF-1+Cx37 group (*p* < 0.05).

### 3.5. Effect of the Combination Treatment of EGF, IGF-1, and Cx37 on GSH, Reactive Oxygen Species (ROS) Level in Bovine Oocytes

As shown in Figure 2A, the GSH level of the EGF+IGF-1+Cx37 group was significantly higher than that of the control group (*p* < 0.05), and lower than that of FSH+LH+EGF+IGF-1+Cx37 group (*p* < 0.05). As shown in Figure 2B, the ROS level in oocytes of the FSH+LH+EGF+IGF-1+Cx37 group was significantly lower than that of EGF+IGF-1+Cx 37 group and control group.

### 3.6. Effect of the Combination Treatment of EGF, IGF-1, and Cx37on Mitochondrial Function in Bovine Oocytes

As shown in Figure 3A, the ATP content in oocytes in the EGF+IGF-1+Cx37 group was significantly higher than that in the control group (*p* < 0.05) and lower than the FSH+LH+EGF+IGF-1+Cx37 group (*p* < 0.05). The same results were found for the expression of the ATP-synthesized genes ATP6 and ATP8 (Figure 3B).

### 3.7. Effect of the Combination Treatment of EGF, IGF-1, and Cx37on the Maturation and Developmental Ability of Bovine Oocytes

As shown in Table 5, the nuclear maturation rate of the EGF+IGF-1+Cx37 group was significantly higher than that of the control group (*p* < 0.05), but lower than that of the FSH+LH+EGF+IGF-1+Cx37 group (*p* < 0.05). At the same time, the cleavage rate and blastocyst rate of the EGF+IGF-1+Cx37 group were significantly higher than those of the control group (*p* < 0.05), but lower than those of the FSH+LH+EGF+IGF-1+Cx37 group (*p* < 0.05).

### 3.8. Effect of the Combination Treatment of EGF, IGF-1, and Cx37on the Quality of IVF Blastocysts

As shown in Figure 4A,B, the total cell number and survival rate after the vitrification of the blastocysts of the EGF+IGF-1+Cx37 group were significantly higher than that of the control group (*p* < 0.05), but lower than that of the FSH+LH+EGF+IGF-1+Cx37 group (*p* < 0.05). Meanwhile, as shown in Figure 4C, the mRNA expression level of IFN-tau, AQP 3, and CTNNBL1 genes in blastocysts of the EGF+IGF-1+Cx37 group was significantly higher than those of the control group but lower than those of the FSH+LH+EGF+IGF-1+Cx37 group. For apoptosis-related genes, the mRNA expression of the anti-apoptosis gene BCL-2 of the EGF+IGF-1+Cx37 group was significantly higher than that of the control group but lower than that of the EGF+IGF-1+Cx37 group, which was contrary to the results in the expression of the pro-apoptosis gene BAX.

## 4. Discussion

During the IVM of bovine oocytes, it has been proven that the treatment with EGF and IGF-1 stimulates cumulus expansion, oxidative metabolism, and nuclear maturation [39], and accelerates oocytes’ meiosis process by increasing the activity of H1 and MAP kinases [40]. Similarly, as shown in Table 2, our experiments showed that when EGF and IGF-1 were both added, the maturation of oocytes and development of their IVF embryos were significantly improved.

Since a high level of gonadotropins affects the maturation and development ability of mammalian oocytes, it is necessary to develop a hormone-free IVM medium for the oocytes culture in vitro. As shown in Table 3, in the absence of FSH or LH, the addition of EGF or IGF-1, especially both EGF and IGF-1, significantly improved the maturation and development ability of bovine oocytes, which confirmed the results in buffalo oocytes [41]. EGF can effectively stimulate a series of cascade reactions and weaken meiosis inhibition, resulting in meiosis of oocytes [18,35]. IGF-1 can promote the proliferation and differentiation of granulosa cells [42] by activating the mitogen-activated protein kinase-signaling cascade [43], and inhibit apoptosis by mediating the phosphoinositide 3-kinase activation of Akt and the inactivation of pro-apoptotic proteins [44], which ultimately promotes oocyte maturation. In the absence of FSH or LH, a combination of EGF and IGF-1 stimulates a cascade of events, including protein synthesis, which eventually generates positive signals for the resumption of meiosis in oocytes [41]. All this evidence contributed to explaining the increased maturation and development of oocytes treated by EGF and IGF-1 without gonadotropins (Table 3).

As shown in Table 4, our results revealed that 25 μg/mL Cx37 efficiently accelerated the subsequent developmental ability of oocytes. Studies have shown that the growth rate of oocytes is proportional to the number of cumulus cells [45]. As a member of the connexin family, Cx37 is present in both cumulus cells and oocytes at all stages of follicular development [46] and has been proven to be essential for the communication between oocyte and cumulus cells via the gap junction [47]. The lack of Cx37 in female mice reduced the meiotic competence of oocytes and resulted in a lack of gap junctions between the oocytes and cumulus cells [48]. The wide gap junction coupling between oocytes and granulosa cells greatly increases the effective plasma membrane surface area, and oocytes can use it to obtain essential nutrients from extracellular sources [49,50], which explained the improved quality and development ability of oocytes treated by Cx37 (Table 4).

As an extension of the cumulus cells, TZP penetrates ZP to contact the oocytes [51], and numerous TZPs develop and contribute to the growth of growing oocytes [45]. TZPs can provide pyruvate, nicotinamide adenine dinucleotide phosphate, and cyclic guanosine monophosphate to oocytes [27]. As shown in Figure 1, our experiments showed that the oocytes treated with EGF, IGF-1, and Cx37 significantly improved the level of TZP in oocytes, which explained the increased maturation and development ability of bovine oocytes.

The GSH level in oocytes is an important factor affecting the quality of IVM of oocytes and subsequent embryonic development [52]. As one kind of antioxidant, GSH content is used as an effective index to measure the cytoplasm maturity of oocytes [53]. GSH can regulate intra-cellular redox balance, protecting cells from oxidative damage [52]. ROS can damage the cell membrane and induce apoptosis, which thereby harms the fertilization potential of oocytes [54]. As shown in Figure 2, our experiments showed that the treatment with EGF, IGF-1, and Cx37 significantly improved the GSH and decreased the ROS level in oocytes, which was due to that that EGF upregulated the expression of antioxidative enzymes (SOD, CAT, and GSH-Px) to alleviate oxidative injury [55,56].

The expression levels of mitochondrial genes affect the quality, fertilization, and embryo development of oocytes [57], and have been used to assess the quality of the oocytes [58]. The ATP-content level is closely associated with the development ability of oocytes in bovine [54], sheep [59], and mice [60]. As shown in Figure 3, the ATP content and mRNA expression of ATP6 and ATP8 were significantly improved by the combination treatment of EGF, IGF-1, and Cx37, and it was due to that that growth factors can increase the mitochondrial membrane potential of bovine oocytes [61] and the ROS can significantly decrease the maturation and ATP content of mouse oocytes [60].

As shown in Table 5, the maturation and development ability of oocytes were significantly improved by the combination treatment of EGF, IGF-1, and Cx37, which was due to the increased level of TZP (Figure 1) and GSH, decreased ROS level (Figure 2), and higher ATP-content level and mRNA expression level (Figure 3). The total cell number per blastocyst [62,63,64,65], IFN-τ expression levels [58,66], and survival rate after vitrification [34] have been described as good indicators to determine the quality of the embryo. AQP3 is a transmembrane channel protein that allows the rapid and passive movement of water, as well as other tiny neutral solutes, across the membrane to improve plasma membrane permeability and blastocyst cavity formation [67]. CTNNBL1 is the marker of compaction and trophectoderm differentiation [58]. *BAX* is a pro-apoptotic protein that leads to cell death, whereas BCL-2 is an anti-apoptotic protein that promotes cell survival, and the ratio of BCL-2 to BAX ratio determines whether a cell survives or undergoes apoptosis [68]. As shown in Figure 4, our experiments showed that EGF, IGF-1, and Cx37 supplementation significantly increased the total number of cells and the expression levels of IFN- Tau, CTNNBL1, AQP3, and BCL-2 genes in their IVF blastocysts, and reduced the expression levels of BAX genes in blastocysts, indicating that the quality of the blastocyst was improved by the combined treatment of EGF, IGF-1, and Cx37.

## 5. Conclusions

In conclusion, our results showed that the supplementation of EGF, IGF-1, and Cx37 in IVM medium of bovine oocytes significantly improved their quality and developmental ability by increasing the TZP, mitochondrial function, and GSH level.

## Figures and Tables

**Figure 1 genes-13-00805-f001:**
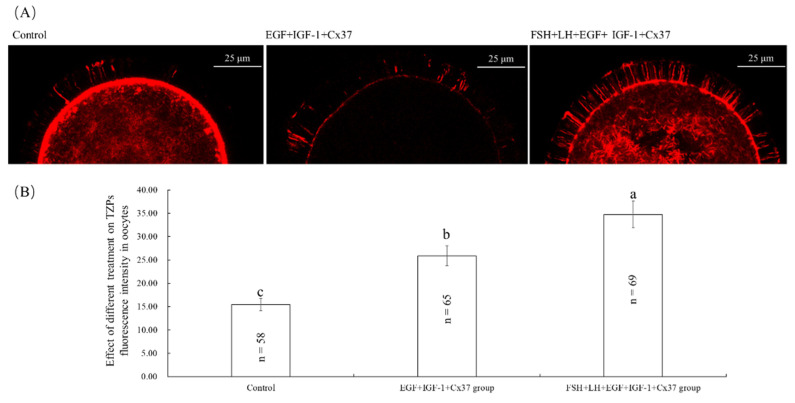
Effect of the combination treatment of EGF, IGF-1, and Cx37 on TZPs in bovine oocytes. (**A**) Representative image staining of TZPs in bovine oocytes. (**B**) Effect of the combination treatment of EGF, IGF-1, and Cx37on TZPs in bovine oocytes. ^a, b, c^ Values with different superscripts indicate significant differences between groups (*p* < 0.05).

**Figure 2 genes-13-00805-f002:**
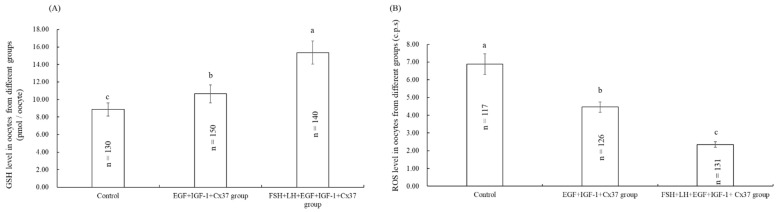
Effect of the combination treatment of EGF, IGF-1, and Cx37 on GSH and ROS level in bovine oocytes. (**A**) Effect of the combination treatment of EGF, IGF-1, and Cx37 on GSH level in bovine oocytes. (**B**) Effect of the combination treatment of EGF, IGF-1, and Cx37 on ROS level in bovine oocytes. ^a, b, c^ Values with different superscripts indicate significant differences between groups (*p* < 0.05).

**Figure 3 genes-13-00805-f003:**
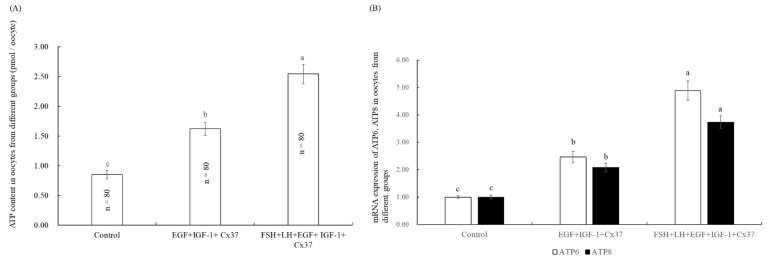
Effect of the combination treatment of EGF, IGF-1, and Cx37on mitochondrial function in bovine oocytes. (**A**) Effect of the combination treatment of EGF, IGF-1, and Cx37 on ATP content in bovine oocytes. (**B**) Effect of the combination treatment of EGF, IGF-1, and Cx37 on mRNA expression of ATP6 and ATP8 level in bovine oocytes. ^a, b, c^ Values with different superscripts indicate significant differences between groups (*p* < 0.05).

**Figure 4 genes-13-00805-f004:**
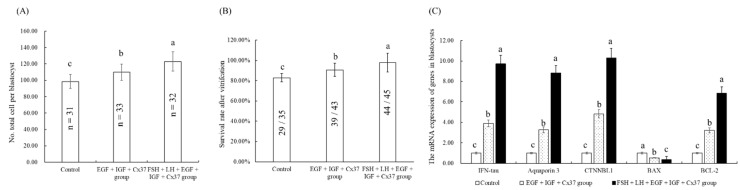
Effect of the combination treatment of EGF, IGF-1, and Cx37on the quality of IVF blastocysts. (**A**) Effect of the combination treatment of EGF, IGF-1, and Cx37 on the total cell number per blastocysts. (**B**) Effect of the combination treatment of EGF, IGF-1, and Cx37on the survival rate of blastocysts after vitrification. (**C**) Effect of the combination treatment of EGF, IGF-1, and Cx37 on the quality-related gene expression in blastocysts. ^a, b, c^ Values with different superscripts indicate significant differences between groups (*p* < 0.05).

**Table 1 genes-13-00805-t001:** Primers used for qRT-PCR of candidate genes in oocytes and blastocysts.

Gene	Primers (5′–3′)	Size (bp)	GenBank Accession No.	Annealing Temperature (°C)
IFN- tau	F: GCTCCAGAAGGATCAGGCTATC	95	AF238611	60
	R: TGTTCCAAGCAGCAGACGAGT			
CTNNBL1	F: GTTCCTGCCTAATGCTGAGTTCC	191	NM_174637.3	60
	R: GGTCCGTAAGCCAAGAATGTCA			
AQP3	F: AACCCTGCTGTGACCTTTGCTA	230	AF123316	60
	R: TTGACCATGTCCAAGTGTCCAG			
BAX	F: TTTCTGACGGCAACTTCAACTG	83	NM_173894.1	60
	R: GGTGCACAGGGCCTTGAG			
BCL-2	F: TCAATTGTCGTGGCATCAAAA	249	NM_001077486.2	60
	R: CCCCCGACACCTGTTAGCTT			
ATP6	F: GAACACCCACTCCACTAATCCCAAT	147	AF493542	60
	R: GTGCAAGTGTAGCTCCTCCGATT			
ATP8	F: CACAATCCAGAACTGACACCAACAA	129	AF493542	60
	R: CGATAAGGGTTACGAGAGGGAGAC			
Β-ACTIN	F: TCCTGGGCATGGAATCCTG	177	NM_173979	60
	R: GGCGCGATGATCTTGATCTTC			

**Table 2 genes-13-00805-t002:** Effect of EGF and IGF-1 on the maturation and development ability of bovine oocytes.

Groups	No. COCs	No. Metaphase (M) II Oocytes	No. Cleavage Embryos	No. Blastocysts
FSH+LH+EGF	187	159 (85.03 ± 5.04%) ^b^	128 (80.50 ± 7.21%) ^b^	39 (30.47 ± 2.71%) ^b^
FSH+LH+IGF-1	194	163 (84.02 ± 4.93%) ^b^	132 (80.98 ± 6.19%) ^b^	41 (31.06 ± 2.63%) ^b^
FSH+LH+EGF+IGF-1	210	194 (92.38 ± 2.18%) ^a^	174 (89.69 ± 4.71%) ^a^	71 (40.80 ± 3.59%) ^a^
Control	291	238 (81.79 ± 1.39%) ^b^	176 (73.95 ± 8.63%) ^b^	51 (28.98 ± 2.45%) ^b^

^a, b^ Values with different superscripts indicate significant differences between groups (*p* < 0.05).

**Table 3 genes-13-00805-t003:** Effect of EGF and IGF-1 on the maturation and development ability of bovine oocytes without gonadotropins.

Groups	No. COCs	No. MII Oocytes	No. Cleavage Embryos	No. Blastocysts
EGF+IGF-1	243	203 (83.54 ± 4.92%) ^a^	143 (70.44 ± 9.83%) ^a^	44 (30.77 ± 3.09%) ^a^
EGF	251	172 (68.53 ± 5.89%) ^b^	93 (54.07 ± 4.34%) ^b^	18 (19.35 ± 1.67%) ^b^
IGF-1	288	200 (69.44 ± 8.44%) ^b^	110 (55.00 ± 5.89%) ^b^	23 (20.91 ± 2.39%) ^b^
-FSH-LH	371	208 (56.06 ± 5.23%) ^c^	94 (45.19 ± 4.47%) ^c^	14 (14.89 ± 1.93%) ^c^
Control	202	162 (80.20 ± 6.81%) ^a^	122 (75.31 ± 5.17%) ^a^	38 (31.15 ± 2.68%) ^a^

^a, b, c^ Values with different superscripts indicate significant differences between groups (*p* < 0.05).

**Table 4 genes-13-00805-t004:** Effect of Cx37 on the maturation and development ability of bovine oocytes.

Groups	No. Cocs	No. MII Oocytes	No. Cleavage Embryos	No. Blastocysts
12.5 μg/mL Cx37	102	84 (82.35 ± 7.82%) ^a^	65 (77.38 ± 6.91%) ^b^	19 (29.23 ± 2.13%) ^b^
25 μg/mL Cx37	116	96 (82.76 ± 7.39%) ^a^	83 (86.46 ± 8.08%) ^a^	32 (38.55 ± 2.59%) ^a^
50 μg/mL Cx37	105	86 (81.90 ± 7.13%) ^a^	68 (79.07 ± 7.12%) ^b^	21 (30.88 ± 2.79%) ^b^
Control	94	76 (80.85 ± 6.34%) ^a^	58 (76.32 ± 4.07%) ^b^	17 (29.31 ± 2.62%) ^b^

^a, b^ Values with different superscripts indicate significant differences between groups (*p* < 0.05).

**Table 5 genes-13-00805-t005:** Effect of the combination treatment of EGF, IGF-1, and Cx37on the maturation and developmental ability of bovine oocytes.

Groups	No. COCs	No. MII Oocytes	No. Cleavage Embryos	No. Blastocysts
EGF+ IGF-1+Cx37	350	306 (87.43 ± 6.86%) ^b^	250 (81.70 ± 7.65%) ^b^	100 (40.00 ± 3.87%) ^b^
FSH+LH+EGF+ IGF-1+Cx37	279	262 (93.91 ± 7.84%) ^a^	237 (90.46 ± 8.28%) ^a^	120 (50.63 ± 4.75%) ^a^
Control	383	310 (80.94 ± 6.64%) ^c^	234 (75.48 ± 6.82%) ^c^	73 (31.20 ± 3.05%) ^c^

^a, b, c^ Values with different superscripts indicate significant differences between groups (*p* < 0.05).

## Data Availability

The data that support the findings of this study are available on request from the corresponding author. The data are not publicly available due to privacy or ethical restrictions.

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
