# Peer review of "Supplementation of EGF, IGF-1, and Connexin 37 in IVM Medium Significantly Improved the Maturation of Bovine Oocytes and Vitrification of Their IVF Blastocysts"

_genes, 2022, doi:10.3390/genes13050805_

Round 1

Reviewer 1 Report

Title: Supplementation of EGF, IGF-1 and Connexin 37 in IVM medium significantly improved the maturation of bovine oocytes and vitrification of their IVF blastocysts

This manuscript is well written and addresses an interesting topic. 

. Abstract: Please define the acronyms IVF and TZP.

. Introduction:

In line 46, define the acronym IVF

In line 50, define the acronym COCs

In line 66, define the acronym TZP

. Tables and figure : define all acronyms, i.e, COCs, MII and TZP

. Figures: inserte "a, b, c: Values with different superscripts indicate significant differences between groups (P < 0.05).

Author Response

Comments to the Author

. Abstract: Please define the acronyms IVF and TZP.

. Introduction:

In line 46, define the acronym IVF

In line 50, define the acronym COCs

In line 66, define the acronym TZP

. Tables and figure: define all acronyms, i.e, COCs, MII and TZP

. Figures: insert "a, b, c: Values with different superscripts indicate significant differences between groups (P < 0.05).

Generally, the paper presents interesting research, however, I recommend the minor revision that should be resolved before its publishing.

Question 1: Abstract: Please define the acronyms IVF and TZP.

ANSWER: Thanks for your suggestions and we had done the revised as follows:

Cytoplasmic maturation events of oocytes and quality of in vitro fertilization (IVF) blastocysts were examined to investigate the relative mechanisms.

The transzonal projection (TZP) intensity, glutathione (GSH) level and mitochondrial function of the EGF+IGF-1+Cx37 group were significantly higher than that of the control group.

Question 2: In line 46, define the acronym IVF

ANSWER: Thanks for your suggestions and we had done the revised as follows:

The volume density of mitochondria and the nuclear volume density of in vitro fertilization (IVF) blastocysts are lower than those in blastocysts obtained in vivo, in contrast to the lipid volume density [14].

Question 3: In line 50, define the acronym COCs

ANSWER: Thanks for your suggestions and we had done the revised as follows:

In vivo, cumulus-oocyte-complexes (COCs) are bathed with follicular fluid, which contains a series of proteins, cytokines, hormones, energy metabolites, steroids, and some undefined factors [17].

Question 4: In line 66, define the acronym TZP.

ANSWER: Thanks for your suggestions and we had done the revised as follows:

However, high doses of FSH may affect oocyte chromatin dynamics and oocyte transcription/translation activities, impel gap connection-mediated communication between cumulus cells and oocytes by reducing transzonal projection (TZP) density [27], and impair oocyte development and embryo production [28].

Question 5: Tables and figure: define all acronyms, i.e, COCs, MII and TZP

Thanks for your suggestions, but we have defined all acronyms in the text, no need to define them again in the figures and tables.

cumulus-oocyte-complexes (COCs)

MII: metaphase (M) II

transzonal projection (TZP)

Question 6: Figures: insert "a, b, c: Values with different superscripts indicate significant differences between groups (P < 0.05).

ANSWER: Thanks for your suggestions and we had added details, all figures had inset "a, b, c: Values with different superscripts indicate significant differences between groups (P < 0.05), please check!

Reviewer 2 Report

Manuscript revision “Supplementation of EGF, IGF-1 and Connexin 37 in IVM medium significantly improved the maturation of bovine oocytes and vitrification of their IVF blastocysts”.
The manuscript is interesting and well written. Even though the idea is not so original, results are convincing.
Some ameliorations are necessary for improving the quality of the paper.
Please, the first-time authors use an abbreviation, they should spell out the full term and put the abbreviation in parentheses, such as cumulus-oocyte complexes (COCs), transzonal projection (TZP), etc.
The results should not be repeated. Since the values are in the tables, they shouldn’t be also in the text. Moreover, the manuscript would be more readable. Please remove them also from the abstract. 
Descriptions should be simplified. Authors could indicate the base medium (199+FBS+estradiol) and then they could distinguish medium with other hormones and medium without hormones.  Please, indicate each group only with added factors: i.e. -FSH-LH+EGF+IGF-1 group will simply be EGF+IGF-1 group, and so on.
67: . And: Moreover,
80: an introductive sentence, explaining the experimental design is needed here.
It has to be clarified what is the control group (what does controls consist of?).
Section “4.10 Experimental design” should be moved and became 4.1.

Author Response

Reply to Reviewer: 2

Comments to the Author

Manuscript revision “Supplementation of EGF, IGF-1 and Connexin 37 in IVM medium significantly improved the maturation of bovine oocytes and vitrification of their IVF blastocysts”.

The manuscript is interesting and well written. Even though the idea is not so original, the results are convincing.

Some ameliorations are necessary for improving the quality of the paper.

Please, the first-time authors use an abbreviation, they should spell out the full term and put the abbreviation in parentheses, such as cumulus-oocyte complexes (COCs), transzonal projection (TZP), etc.

The results should not be repeated. Since the values are in the tables, they shouldn’t be also in the text. Moreover, the manuscript would be more readable. Please remove them also from the abstract.

Descriptions should be simplified. Authors could indicate the base medium (199+FBS+estradiol) and then they could distinguish between medium with other hormones and medium without hormones.  Please, indicate each group only with added factors: i.e. -FSH-LH+EGF+IGF-1 group will simply be EGF+IGF-1 group, and so on.

67: .And: Moreover,

80: an introductive sentence, explaining the experimental design is needed here.

It has to be clarified what is the control group (what does controls consist of?).

Section “4.10 Experimental design” should be moved and become 4.1.

Question 1: Please, the first-time authors use an abbreviation, they should spell out the full term and put the abbreviation in parentheses, such as cumulus-oocyte complexes (COCs), transzonal projection (TZP), etc

ANSWER: Thanks for your suggestions we had written the full spelling of the corresponding noun in place and put the abbreviation in parentheses, please check!

 Question 2: The results should not be repeated. Since the values are in the tables, they shouldn’t be also in the text. Moreover, the manuscript would be more readable. Please remove them also from the abstract.

ANSWER: Thanks for your suggestions and we had revised it, please check!

Question 3: Descriptions should be simplified. Authors could indicate the base medium (199+FBS+estradiol) and then they could distinguish between medium with other hormones and medium without hormones.  Please, indicate each group only with added factors: i.e. -FSH-LH+EGF+IGF-1 group will simply be EGF+IGF-1 group, and so on.

ANSWER: Thanks for your suggestions and we had revised it, please check!

Question 4: 67:And: Moreover,

ANSWER: Thanks for your suggestions and we had revised it, please check!

Question 5: 80: an introductive sentence, explaining the experimental design is needed here.

It has to be clarified what is the control group (what does controls consist of?).

Section “4.10 Experimental design” should be moved and became 4.1.

ANSWER: Thanks for your suggestions and we had revised it, please check!

The Experimental design has been introduced in detail in “4. Materials and Methods”, and the control group has also been described in detail. Meanwhile, the content of the original “4.10 Experimental design” has been integrated into 4.1.

Now the part “4.1. IVM of bovine oocytes” is as follows

Bovine ovaries were collected from local slaughterhouse, and transported to the laboratory within 2 h in a physiological saline solution with penicillin and streptomycin. Follicles with a diameter of 2-8mm were selected to collect cumulus-oocyte complexes (COCs), and only those with more than 3 layers of compact cumulus cells were used for the experiment. For IVM, 50 COCs were cultured in each well containing 750 μL IVM medium of 4-well dishes at 38.5℃ with 5 % CO2 for 22-24 h. The basic IVM medium contained medium 199 (Gibco BRL, Carlsbad, CA, USA) supplemented with10% (v/v) fetal bovine serum (FBS, Hyclone; Gibco BRL), 10 μg/mL estradiol.

According to the experiment design, the oocytes were cultured in different IVM medium as below.

For the control group, the IVM meidum contained medium 199 (Gibco BRL, Carlsbad, CA, USA) supplemented with10% (v/v) fetal bovine serum (FBS, Hyclone; Gibco BRL), 5 IU/mL FSH, 10 IU/mL LH, 10 μg/mL estradiol.

In the first experiment, 100 ng/ml IGF-1 or 50 ng/ml EGF were added into the basic IVM medium of bovine COCs in the presence of FSH and LH, and the maturation and development ability of bovine oocytes were examined.

In the second experiment, 100 ng/ml IGF-1 or 50 ng/ml EGF were added into the basic IVM medium of bovine COCs without FSH and LH, and the maturation and development ability of bovine oocytes were examined.

In the third experiment, different concentrations of Cx37 (0, 12.5μg/mL, 25μg/mL, 50μg/mL) were added to the basic IVM medium of bovine COCs in the presence of FSH and LH, and nuclear maturation and development, and the optimal Cx37 concentration were examined.

Finally, 100 ng/ml IGF-1, 50 ng/ml EGF and 25μg/mL Cx37 were added together into the basic IVM medium of oocytes with or without FSH and LH, and cytoplasmic events of bovine oocytes and quality of their IVF blastocysts were examined.
